# Mpox Infection and Endocrine Health: Bridging the Knowledge Gap

**DOI:** 10.3390/medicina61050899

**Published:** 2025-05-15

**Authors:** Christos Savvidis, Manfredi Rizzo, Ioannis Ilias

**Affiliations:** 1Department of Endocrinology, Hippokration General Hospital, 11527 Athens, Greece; csendo@yahoo.gr; 2Department of Health Promotion, Mother and Child Care, Internal Medicine and Medical Specialties (Promise), School of Medicine, University of Palermo, 90127 Palermo, Italy; manfredi.rizzo@unipa.it

**Keywords:** Mpox, Monkeypox virus, endocrine dysfunction, subacute thyroiditis, adrenal insufficiency, reproductive health

## Abstract

Mpox (MPX), caused by the Monkeypox virus (MPXV), is a zoonotic orthopoxvirus infection with increasing global relevance due to sustained human-to-human transmission. While primarily known for cutaneous and systemic involvement, emerging evidence suggests that MPX may also disrupt endocrine function. This narrative review aims to synthesize current clinical, experimental, and epidemiological findings on MPX-related endocrine complications. We explore the potential impact of MPXV on the thyroid, adrenal glands, and gonads, and discuss the underlying mechanisms, clinical manifestations, and implications for patient management. MPX has been implicated in viral-induced subacute thyroiditis, with cases exhibiting thyrotoxicosis followed by hypothyroidism, likely mediated by direct viral infiltration or immune dysregulation. Additionally, MPX may contribute to adrenal insufficiency through viral invasion, immune-mediated destruction, or hypothalamic–pituitary–adrenal (HPA) axis dysfunction, exacerbating metabolic and inflammatory complications. MPXV’s persistence in testicular tissue raises concerns about reproductive health, with potential implications for fertility, hormone production, and viral transmission. The virus may also modulate host steroid pathways through interactions with glucocorticoid, androgen, and estrogen receptors, influencing immune responses and disease severity. Given these findings, clinicians should maintain vigilance for endocrine dysfunction in MPX patients, particularly in immunocompromised individuals. The role of steroid therapy in MPX remains complex, requiring careful balancing of its anti-inflammatory benefits against potential risks of viral persistence and immune suppression. Further research is essential to clarify MPX’s endocrine impact and optimize management strategies.

## 1. Introduction

Mpox (MPX), previously known as Monkeypox, is a zoonotic disease caused by the Monkeypox virus (MPXV), a double-stranded DNA virus belonging to the Orthopoxvirus genus within the Poxviridae family [1]. MPXV primarily circulates among wild animals, particularly rodents, with occasional spillover to humans leading to outbreaks. Historically, Mpox was confined to Central and West Africa. However, recent sustained human-to-human transmission has resulted in its emergence in non-endemic regions, significantly elevating global health concerns about the potential for widespread outbreaks [2,3,4,5].

The global importance of Mpox was underscored when the World Health Organization declared it a Public Health Emergency of International Concern. In 2024, the Democratic Republic of the Congo (DRC) reported over 16,800 cases and 500 deaths, with children under 15 comprising nearly 70% of cases and 85% of deaths, highlighting the disproportionate burden on pediatric populations [5]. The Mpox outbreak poses several pressing public health and systemic challenges. In many low-resource settings, limited diagnostic capacity and weak surveillance systems hinder timely detection and containment efforts, exacerbating the spread of the disease [6]. Additionally, there are stark vaccine inequities: while high-income countries rapidly secured the majority of available Mpox vaccine doses, endemic African nations continue to face shortages due to prohibitive costs and constrained supply chains [7]. These disparities highlight the urgent need for a globally coordinated strategy to ensure equitable access to prevention and control measures. Additionally, the economic burden is substantial; one study estimated the direct healthcare expenses associated with quarantining Monkeypox patients to total USD 1,791,172, averaging approximately USD 3755 per patient [8].

Viral infections are known to have endocrine repercussions [9], yet MPX’s impact on the endocrine system has been the subject of few reports. The aim of the present narrative review is to assess the reported and tentative endocrine repercussions of MPX.

## 2. MPX and Thyroid Function

Emerging evidence suggests that MPX infection may contribute to thyroid dysfunction, particularly subacute thyroiditis, an inflammatory disorder frequently associated with viral infections such as influenza, enteroviruses, Epstein–Barr virus, cytomegalovirus, and SARS-CoV-2 [9]. While the role of MPX in thyroid dysfunction is still under investigation, recent clinical and experimental findings indicate its potential involvement in thyroid inflammation and immune-mediated thyroid damage.

One of the most significant clinical links between MPX and subacute thyroiditis is a reported case of a 51-year-old male with well-controlled HIV who developed thyrotoxicosis following MPX infection. The patient initially experienced fever, night sweats, weight loss, and neck discomfort approximately three weeks after recovering from MPX-related skin lesions. Laboratory tests revealed suppressed thyrotropin (TSH) levels and elevated free triiodothyronine (T3) and thyroxine (T4), consistent with thyrotoxicosis, while negative anti-thyroid antibodies ruled out autoimmune thyroid disease. Treatment with prednisolone resulted in rapid symptom resolution within 24 h, yet the patient later transitioned into a hypothyroid phase, ultimately requiring levothyroxine therapy to maintain normal thyroid function [10].

Animal studies further support the association between MPX and thyroid inflammation. Infected primates have exhibited thyroid inflammation, suggesting a viral etiology for thyroiditis [11]. Findings from experimental studies indicate that viral infiltration of thyroid follicles may contribute to immune-mediated thyroid destruction, a characteristic feature of subacute thyroiditis [12]. The mechanisms by which MPX induces thyroiditis remain unclear, but two primary hypotheses have been proposed. The first suggests that MPXV directly invades thyroid follicular cells, leading to inflammation and cellular destruction, similar to the process observed in SARS-CoV-2-induced thyroiditis [13]. This hypothesis is further supported by histopathological studies, which have demonstrated necrosis and immune cell infiltration in the thyroid tissue of infected animals [14]. The second hypothesis focuses on an immune-mediated mechanism, in which MPX infection triggers immune dysregulation, leading to an inflammatory response directed at thyroid tissue. Similar immune-driven mechanisms have been observed in post-viral thyroiditis, where viral infections provoke an autoimmune-like inflammatory attack, resulting in an initial hyperthyroid phase followed by transient or permanent hypothyroidism [15].

The clinical course of MPX-associated thyroiditis appears to follow a progression similar to classic subacute thyroiditis. Initially, the destruction of thyroid follicles leads to the excessive release of stored thyroid hormones, causing thyrotoxicosis. As inflammation subsides, thyroid function temporarily normalizes in a euthyroid phase, but some patients eventually develop hypothyroidism, which may be transient or permanent, depending on the extent of follicular damage. Current treatment strategies focus on symptom management, with beta-blockers such as propranolol used to control thyrotoxic symptoms, while corticosteroids, particularly prednisolone, have demonstrated rapid effectiveness in reducing thyroid inflammation in reported cases. However, long-term monitoring remains crucial, as some individuals may progress to persistent hypothyroidism requiring ongoing levothyroxine therapy. Clinicians should maintain a high index of suspicion for thyroid dysfunction in MPX patients, especially in those presenting with unexplained fatigue, weight fluctuations, neck pain, or tachycardia, with particular attention to immunocompromised individuals, including those living with HIV. While case reports suggest subacute thyroiditis in Mpox, data from a global cohort provide limited direct evidence of thyroid dysfunction. In the international series by Thornhill et al. (2022) [16], thyroid-specific symptoms were not documented, though systemic inflammatory symptoms such as fever (62%) and lymphadenopathy (56%) were common. These findings may reflect subclinical endocrine involvement. Prospective endocrine monitoring in similar cohorts would help clarify the true incidence of thyroid involvement.

## 3. MPX and the Adrenals/Other Endocrine Glands

Beyond thyroid dysfunction, emerging evidence suggests that MPX may have broader endocrine effects, including potential impact on adrenal function. Such a disruption could contribute to metabolic disturbances and immune dysregulation, further complicating the clinical management of affected individuals [15]. The adrenal glands play a critical role in immune modulation, stress adaptation, and metabolic regulation through the secretion of glucocorticoids and mineralocorticoids. Viral infections have been implicated in adrenal insufficiency (AI), particularly in cases of systemic inflammation and immune activation, which can disrupt adrenal homeostasis [17]. The development of AI in the context of MPX infection may arise due to direct viral invasion of adrenal tissue, immune-mediated destruction, or hypothalamic–pituitary–adrenal (HPA) axis dysregulation.

Patients with adrenal insufficiency who require glucocorticoid replacement therapy already face heightened risks due to impaired stress responses and metabolic instability. The added burden of systemic viral infections, such as MPX, could further exacerbate the risk of adrenal crises, increase inflammatory burden, and worsen metabolic complications [18]. Studies have shown that inadequate or excessive glucocorticoid therapy in AI patients contributes to cardiovascular and metabolic disturbances, a concern that may be amplified in individuals infected with MPX [19].

Glucocorticoid metabolism is precisely regulated by enzymes such as 11β-hydroxysteroid dehydrogenase (11β-HSD), which control cortisol activation and inactivation within tissues. Alterations in 11β-HSD activity are known to influence tissue-specific cortisol exposure, particularly in cases of chronic inflammation and endocrine dysfunction [20]. Given that MPX triggers systemic inflammation, it is possible that the infection perturbs glucocorticoid homeostasis, leading to increased cortisol activation and metabolic dysregulation [21].

The immune response to MPX involves profound cytokine activation, which can interfere with adrenal and other endocrine functions. Though animal models have hinted at adrenal suppression due to orthopoxviral infection, real-world data remain sparse. In the UK-based observational study by Adler et al. (2022) [22], adrenal insufficiency was not clinically confirmed, but prolonged illness (hospital stays of up to 39 days) and fatigue suggest a possible adrenal axis dysregulation. Moreover, systemic features like low mood, malaise, and liver enzyme alterations may reflect broader endocrine-immune interaction that warrants investigation.

Persistent inflammation is known to reduce 11β-HSD2 activity, leading to prolonged glucocorticoid exposure in mineralocorticoid-sensitive tissues such as the kidneys. This dysregulation may contribute to hypertension and cardiovascular risk, which is already a major concern in patients with endocrine disorders and chronic inflammation [23].

Given the complex interplay between adrenal function, glucocorticoid metabolism, and immune regulation, the use of steroids in MPX management raises important clinical considerations. While steroids are widely used for their anti-inflammatory and immunosuppressive effects, they may modulate disease progression through interactions with hormone receptors and metabolic pathways. Understanding the potential impact of MPX on endocrine homeostasis is crucial for optimizing treatment strategies and preventing long-term complications in affected individuals.

## 4. MPX and the Gonads: Clinical and Pathophysiological Considerations

Growing evidence indicates that MPX may affect reproductive health. Studies in non-human primates have documented MPXV persistence in testicular tissue during the convalescent phase, accompanied by inflammation and necrosis, raising concerns about potential impacts on fertility, hormone production, and sexual transmission [16]. The detection of MPXV in semen and reproductive tissues suggests a risk of viral-induced endocrine disruption, particularly in individuals with pre-existing hormonal imbalances or those undergoing fertility treatment. These findings highlight the need for further research into the long-term reproductive consequences of MPX, especially in immunocompromised individuals, where the risk of gonadal complications may be heightened. While direct evidence of MPX-induced gonadal dysfunction in humans remains limited, emerging data underscore the importance of continued investigation.

The pathophysiology of MPXV extends beyond cutaneous and mucosal involvement, making its effects on reproductive function a subject of ongoing research [16]. The detection of MPXV DNA in semen has introduced questions about a possible route of viral dissemination, though its significance in sexual transmission and reproductive health remains uncertain [24]. The 2022 MPX outbreak revealed novel clinical presentations, including anogenital lesions, suggesting a strong sexual transmission component [25].

Findings from non-human primates provide further insight into MPX’s potential effects on the gonads. MPXV persistence in testicular tissues of macaques, along with associated inflammation and necrosis, raises concerns about viral-mediated testicular damage [26]. Viral infections such as mumps and Zika virus have been linked to fibrosis and impaired spermatogenesis, and while similar mechanisms may be relevant to MPX, definitive evidence of MPXV-induced infertility in humans is lacking [26].

The presence of MPXV DNA in semen has prompted speculation about viral persistence in reproductive tissues, but whether this translates into prolonged infectivity or reproductive consequences remains unclear [24]. Previous studies on viruses like Zika and Ebola have demonstrated extended viral shedding in semen, suggesting that a similar mechanism could be at play with MPXV, though additional research is required to confirm this possibility [24].

Immunocompromised individuals, particularly those living with HIV, may be at greater risk of prolonged viral persistence and MPX-related complications [16]. Chronic inflammation in this population could contribute to testicular damage and endocrine disruptions, emphasizing the need for closer monitoring of gonadal function during and after infection [25]. The interplay between MPXV and other sexually transmitted infections remains an area of concern, particularly in high-risk populations, though direct evidence of sexually transmitted infection (STI)-related gonadal dysfunction in MPX patients is currently limited [25].

While the impact of MPXV on the hypothalamic–pituitary–gonadal (HPG) axis has yet to be established, systemic inflammation triggered by viral infections is known to disrupt hormonal balance, potentially affecting testosterone levels, libido, and muscle function [27]. Although data on MPX-induced hypogonadism remain sparse, ongoing research is needed to determine whether MPXV influences endocrine function in a manner similar to other viral infections [27].

Given the high prevalence of anogenital lesions and the detection of MPXV in reproductive fluids, concerns regarding fertility and reproductive health continue to grow [16]. Longitudinal studies assessing testicular function in MPX survivors, particularly those with severe infection or prolonged viral shedding, are essential to clarify potential long-term effects [28]. The need for clinical vigilance regarding MPX-related reproductive complications has been previously emphasized, though contemporary data on testicular outcomes remain limited [29].

Healthcare professionals should remain attentive to possible gonadal complications of MPX, particularly in high-risk populations such as MSM (men who have sex with men) individuals and those living with HIV [25]. Early identification of symptoms like testicular pain, swelling, or hormonal imbalances may facilitate timely intervention and risk mitigation [24]. Clinical implications for gonadal function are of particular concern given the virus’s mucocutaneous and systemic involvement. In Thornhill et al.’s [16] cohort, genital lesions were seen in over 73% of cases, and seminal fluid tested positive for viral DNA in 91% (29/32 tested). While viral shedding in semen does not confirm endocrine disruption, it emphasizes the reproductive tract’s vulnerability. Future studies should evaluate impacts on testosterone, LH, FSH, and fertility parameters post-infection [21]. Given the uncertainties surrounding MPXV’s impact on fertility, further studies are required to determine whether MPX could lead to transient or permanent gonadal dysfunction [28]. Research focusing on viral persistence in reproductive tissues, potential effects on spermatogenesis, and broader endocrine implications will be critical in shaping long-term management strategies for affected individuals [26].

## 5. Interaction of MPX with Hormonal Receptors and Hormonal Metabolism

Steroids are widely used for their anti-inflammatory and immunosuppressive effects, making their role in MPX infection particularly relevant. The interaction between MPXV and hormone receptors, including glucocorticoid receptors (GRs), androgen receptors (ARs), and estrogen receptors (ERs), raises important clinical considerations [30].

MPXV primarily enters through cutaneous lesions and mucosal surfaces, infecting antigen-presenting cells before disseminating systemically [31]. The immune system responds with innate and adaptive mechanisms, where cytokines such as TNF-α, interleukins, and interferons play key roles [31]. However, MPXV has evolved immune evasion strategies, including interferon signaling inhibition and the modulation of apoptotic pathways, which may impact host immune responses and disease progression [32].

Glucocorticoids (GCs) suppress pro-inflammatory cytokines and inhibit T-cell activation, which may modulate immune responses against MPXV. Research on other orthopoxviruses suggests that steroid-induced immunosuppression can increase susceptibility to severe disease, though direct evidence in MPX remains under investigation [33].

MPXV encodes the SalF7L protein, which shares structural similarity to human 3β-hydroxysteroid dehydrogenase (3β-HSD). This enzyme is involved in steroid metabolism, converting pregnenolone to progesterone, a precursor for glucocorticoids, mineralocorticoids, and androgens [34]. Functional studies suggest that SalF7L exhibits steroidogenic activity, potentially aiding immune evasion by disrupting endocrine homeostasis. In animal models, the deletion of SalF7L led to reduced virulence, highlighting its role in MPXV pathogenesis. Given its ability to manipulate host steroid pathways, further research is needed to determine whether MPXV-induced steroidogenic activity contributes to observed hormonal imbalances in individuals infected with MPX.

Beyond glucocorticoid signaling, ARs and ERs also contribute to immune function and viral pathogenesis. The MAPK signaling pathway, which interacts with hormone receptors, plays a role in MPXV replication and immune modulation, linking hormonal activity to disease severity [35]. Studies on smallpox and other orthopoxviruses suggest that sex hormones influence disease progression, with androgens potentially exacerbating immune suppression, while estrogens may provide protective effects [31].

Emerging evidence indicates that AR signaling could contribute to sex-based disparities in MPX infection and severity. Gene ontology and KEGG pathway analyses have shown that AR signaling modulates host immune responses and inflammation, potentially influencing MPXV pathogenesis [30]. This aligns with epidemiological trends, where MPX cases have been disproportionately reported in males, particularly among individuals with prolonged close contact exposures. MPXV may exploit AR-related pathways to enhance viral replication and immune evasion, but further research is required to clarify these interactions (Figure 1).

## 6. Clinical Implications and Risks of Steroid Therapy in MPX

The use of steroids in MPXV-infected patients presents a clinical dilemma, as their effects on inflammation and immune suppression may influence disease progression. In severe MPX cases characterized by excessive inflammatory responses, steroids may help reduce immune-mediated tissue damage and prevent complications. However, their use in early-stage infections could theoretically prolong viremia and impact viral clearance, a phenomenon observed in other viral infections [33].

Patients on long-term steroid therapy, such as those with autoimmune diseases or undergoing cancer treatment, may be at higher risk of severe MPX outcomes due to compromised immune surveillance and reduced antiviral defenses [31]. Additionally, steroid-induced modulation of hormone receptor signaling may contribute to variations in symptom severity, including rash distribution, lymphadenopathy, and respiratory complications, as seen in immune-related dermatological conditions [36].

Further research is needed to clarify MPX’s interactions with the endocrine system and immune response, particularly regarding the impact of GR polymorphisms on disease progression and treatment outcomes. While variations in GR expression and function have been shown to influence immune responses in other infections, their role in MPX pathogenesis remains speculative and requires further investigation [35]. Another important area of study is the potential use of selective glucocorticoid receptor modulators (SGRMs), which may allow for inflammation control while preserving antiviral immunity [31].

Sex-based differences in MPX disease progression also warrant investigation, particularly regarding androgen and estrogen receptor activity and their influence on immune responses. Studies on orthopoxviruses suggest that androgens may contribute to immune suppression, whereas estrogens may have protective effects, potentially explaining differences in MPX severity across sexes [37].

Additionally, strategies to modulate the immune response in MPX patients without inducing excessive immunosuppression should be explored. Identifying novel immunomodulatory approaches that control cytokine storms while maintaining effective antiviral defenses could significantly improve MPX management and patient outcomes [38].

Given the potential risks associated with steroid therapy in MPX treatment, clinicians must carefully balance their benefits in controlling inflammation with the potential for worsening viral persistence or immunosuppression. Future research should focus on individualized treatment strategies that optimize immune modulation while minimizing adverse effects in individuals infected with MPX.

## 7. Conclusions

As Mpox (MPX) continues to emerge as a globally relevant infectious disease, its potential to disrupt endocrine function warrants heightened attention in both clinical and public health contexts. Evidence suggesting MPX-induced thyroiditis, adrenal insufficiency, and gonadal involvement emphasizes the virus’s capacity to affect multiple endocrine axes, with implications that extend beyond acute infection. These key endocrine effects of MPX are summarized in Table 1.

These findings should encourage clinicians to adopt a more proactive approach in assessing endocrine health among MPX patients, particularly those who are immunocompromised or experience prolonged symptoms. Routine screening for thyroid, adrenal, and gonadal function—using targeted hormonal panels and symptom-based monitoring—may help identify subclinical dysfunctions that otherwise go undetected. Endocrine evaluation could be integrated into diagnostic protocols for MPX, particularly in patients with fatigue, mood changes, electrolyte imbalances, or reproductive concerns.

From a treatment perspective, awareness of MPX’s endocrine effects can influence therapeutic decisions, especially regarding corticosteroid use. Given the delicate balance between immunosuppression and endocrine modulation, clinicians must weigh the risks of delayed viral clearance against the need for inflammation control. The potential role of selective glucocorticoid receptor modulators (SGRMs) and personalized endocrine-based interventions should be explored further.

Public health policymakers should also recognize the broader implications. Incorporating endocrine endpoints into MPX surveillance systems and post-infection registries could enhance the early detection of long-term complications. Education campaigns targeting both healthcare providers and high-risk populations should highlight possible endocrine symptoms of MPX to facilitate timely diagnosis and management.

Ultimately, a better understanding of MPX–endocrine interactions will enhance patient care, inform targeted therapeutic strategies, and shape future guidelines for managing viral infections with systemic consequences. Multidisciplinary collaboration—bridging infectious disease, endocrinology, and epidemiology—is essential to fully elucidate and address these complex interactions.

## 8. Limitations and Future Directions

### 8.1. Limitations

Despite growing interest in the endocrine manifestations of Mpox, current understanding remains limited by the nature of available data. Much of the evidence to date is derived from case reports and preclinical animal studies, which, while informative, lack the robustness of prospective cohort investigations [10,23]. Most published clinical datasets do not include detailed hormonal profiles or standardized endocrine assessments, making it difficult to determine the true prevalence and severity of dysfunction across different axes. Furthermore, the causal relationship between Mpox infection and endocrine abnormalities is not yet well defined. Many reported symptoms could be indirect consequences of systemic inflammation or pre-existing health conditions rather than direct viral impact. Although large-scale studies such as those by Thornhill et al. and Adler et al. provide valuable epidemiological context, they do not specifically address hormonal outcomes, leaving a significant knowledge gap in this area [21,26].

### 8.2. Future Research Directions

To address these limitations, future research should prioritize the establishment of endocrine follow-up registries for individuals recovering from Mpox. These registries should include serial assessments of hormone levels related to the thyroid, adrenal, and gonadal axes, and aim to detect both overt and subclinical dysfunction. Longitudinal cohort studies are also needed to monitor the recovery of the hypothalamic–pituitary–adrenal (HPA) and hypothalamic–pituitary–gonadal (HPG) axes, particularly in individuals with moderate to severe disease or those with prolonged systemic symptoms. Another promising avenue involves the development of immunoendocrine biomarkers, which could help identify early signs of endocrine disruption and facilitate timely intervention. Furthermore, interdisciplinary collaboration between infectious disease specialists, endocrinologists, and epidemiologists will be essential to better understand the long-term outcomes of Mpox-related endocrine alterations. Finally, integrating endocrine endpoints into Mpox surveillance systems and therapeutic trials will provide more definitive insights into the viral impact on hormonal homeostasis and guide evidence-based interventions.

## Figures and Tables

**Figure 1 medicina-61-00899-f001:**
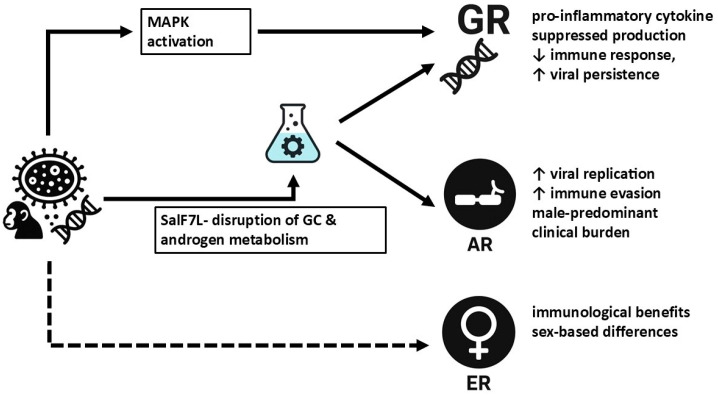
Mechanisms of Mpox–hormonal receptor interactions: The Monkeypox virus (MPXV) modulates host endocrine signaling by interacting with nuclear hormone receptors, notably glucocorticoid receptors (GRs), androgen receptors (ARs), and estrogen receptors (ERs), with implications for immune regulation and disease severity. GR engagement normally suppresses pro-inflammatory cytokine production, but during MPXV infection, MAPK pathway activation may impair GR function, reducing its anti-inflammatory efficacy and potentially facilitating viral persistence. MPXV may also exploit AR signaling to enhance viral replication and immune evasion, contributing to the male-predominant clinical burden observed in outbreaks. A key viral factor, the SalF7L protein, exhibits structural similarity to human 3β-hydroxysteroid dehydrogenase (3β-HSD), enabling disruption of steroid metabolism and endocrine homeostasis by interfering with androgen and glucocorticoid biosynthesis. While ERs are not directly targeted (square dot line) by MPXV, estrogen-related pathways may offer immunological benefits, possibly explaining sex-based differences in clinical outcomes. These interactions underscore MPXV’s capacity to manipulate host hormonal networks and highlight the importance of individualized approaches in managing inflammation and immune responses, particularly in the context of steroid therapy. MPXV: Monkeypox virus, GR: glucocorticoid receptor, AR: androgen receptor, ER: estrogen receptor, MAPK: Mitogen-Activated Protein Kinase, 3β-HSD: 3β-hydroxysteroid dehydrogenase, SalF7L: MPXV-encoded viral protein with 3β-HSD-like activity.

**Table 1 medicina-61-00899-t001:** Summary of the impact of Mpox (MPX) on the endocrine system.

Impact	Findings	Mechanisms	Clinical Implications
Thyroid	MPX linked to subacute thyroiditis, leading to thyrotoxicosis or hypothyroidism	Viral infiltration; immune-mediated damage	Symptoms: fever, weight loss, neck pain Treatment: corticosteroids, levothyroxine
Adrenal	Viral infections have been implicated in adrenal insufficiency	Viral invasion; HPA axis dysregulation	Possible risk of adrenal insufficiency, requiring glucocorticoid therapy
Reproductive (Men)	MPXV persistence in testes; may affect fertility, hormone production	Testicular inflammation; necrosis	Potential sperm quality reduction, hormonal imbalances Further research needed
Hormonal Receptors	MPXV may alter steroid receptor activity	Modulation of glucocorticoid, androgen, and estrogen signaling/metabolism pathways	Possible sex-based differences in severity of MPX
Steroid Therapy	Steroids help inflammation but may prolong MPXV viremia	Interaction with glucocorticoid receptors	Need for balance between benefit and harm needed; selective glucocorticoid receptor modulators (SGRMs) may help

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
