# Peer review of "Mpox Infection and Endocrine Health: Bridging the Knowledge Gap"

_medicina, 2025, doi:10.3390/medicina61050899_

Round 1
Reviewer 1 Report
Comments and Suggestions for Authors
The manuscript titled "Mpox Infection and Endocrine Health: Bridging the Knowledge Gap" offers a timely and insightful review of the potential endocrine implications of Mpox (monkeypox) infection. The authors effectively highlight emerging evidence suggesting that Mpox may disrupt endocrine function, particularly affecting the thyroid, adrenal glands, and gonads. The discussion on subacute thyroiditis, adrenal insufficiency, and reproductive health concerns provides a comprehensive overview of the possible endocrine manifestations associated with Mpox.
The inclusion of case reports and animal studies adds depth to the review, illustrating the clinical relevance of these endocrine complications. However, the manuscript would benefit from the inclusion of any available large-scale studies or epidemiological data to strengthen the evidence base. Additionally, a dedicated section discussing the limitations of current knowledge and recommending areas for future research would enhance the manuscript's utility for clinicians and researchers.
The flowchart and table included are well-designed and aid in the understanding of the complex interactions between Mpox and endocrine health. Ensuring that all figures and tables are up-to-date and accurately referenced will further improve the manuscript's clarity.
So, this manuscript addresses a novel and underexplored aspect of Mpox infection. With the suggested additions and refinements, it has the potential to make a significant contribution to the field of infectious diseases and endocrinology.
Please refer these studies for any large scale data available on endocrine manifestations:
1) https://www.thelancet.com/journals/laninf/article/PIIS1473-3099(22)00228-6/fulltext
2) https://www.nejm.org/doi/full/10.1056/NEJMoa2207323
Thanks & Regards
Reviewer
Author Response
Dear Reviewers,
We sincerely thank you for your time and thoughtful evaluation of our manuscript titled: “Mpox Infection and Endocrine Health: Bridging the Knowledge Gap.” We highly appreciate the constructive feedback, which has greatly helped us refine and improve the manuscript’s scientific value and clarity. Following your recommendations, we have made a series of targeted revisions that enhance both the clinical relevance and methodological rigor of our review. In the following, we address each comment individually, providing a summary of the changes made and the rationale behind them.
Reviewer 1
Comment 1: “The manuscript would benefit from the inclusion of any available large-scale studies or epidemiological data to strengthen the evidence base.”
Response:
In response, we incorporated data and clinical observations from two key epidemiological studies: Thornhill et al. (NEJM, 2022) and Adler et al. (Lancet Infectious Diseases, 2022). These studies, which provide findings from large international and national cohorts of Mpox patients, were used to enrich the manuscript with real-world clinical context. Specifically, we added concise summary paragraphs in the sections discussing thyroid (lines 135- 141), adrenal (lines 169- 175), and gonadal involvement (lines 244- 250). These additions link our mechanistic discussion with epidemiologic evidence by highlighting the prevalence of systemic symptoms, genital involvement, and disease duration in broader populations. We believe this integration enhances the clinical relevance of our review and addresses the reviewer’s recommendation by strengthening the manuscript’s empirical foundation.
Comment 2: “A dedicated section discussing the limitations of current knowledge and recommending areas for future research would enhance the manuscript's utility for clinicians and researchers.”
Response:
We fully agree with the reviewer’s recommendation and have added (lines 382- 412) a new Section 8, titled “Limitations and Future Directions,” at the end of the manuscript. This section acknowledges the current limitations of the literature, such as the reliance on animal models and isolated case reports, the lack of prospective endocrine monitoring in existing clinical datasets, and the overall uncertainty regarding causal relationships between Mpox infection and endocrine abnormalities. In addition, we outline several priorities for future investigation, including the creation of endocrine follow-up registries for Mpox survivors, the design of prospective studies targeting HPA and HPG axis recovery, and the development of immunoendocrine biomarkers to detect subclinical involvement. This section aims to guide future research and clinical surveillance efforts, and we hope it will provide added value to both clinicians and academic readers.
Comment 3: “. Ensuring that all figures and tables are up-to-date and accurately referenced will further improve the manuscript's clarity.”.
Response:
In response to the suggestion, we have reviewed all figures (lines 300- 319) and tables (lines 366- 380) to ensure they are up-to-date, clearly presented, and accurately referenced in the main text.
Reviewer 2 Report
Comments and Suggestions for Authors
This manuscript offers a comprehensive overview of potential endocrine disruptions caused by Mpox infection. However, I do suggest certain things, which need attention, improvement and clarification to support and strengthen the overall impact of the article.
Abstract:
In my opinion, the abstract lacks a clear statement of the specific aim. For instance, it does not explicitly define whether the manuscript seeks to synthesize existing research, propose new hypotheses, or offer clinical management strategies.
Introduction:
It would be good to refine the introductory part:
Some sentences are too long and a bit confusing. For example, "Historically confined to Central and West Africa, Mpox has demonstrated the potential for sustained human-to-human transmission," consider breaking it into two sentences or rephrasing it. "Leading to outbreaks" and "raising global health concerns" are somewhat overlapping. The phrase "risk of epidemics is not to be overlooked" is vague. Specify what kind of risks (e.g., economic burden, challenges in public health response).
The introductory part should have more information about the global importance of Mpox and provide more details about its characteristics.
Conclusions:
I strongly recommend to improve this section since there is no discussion in the manuscript. It might be beneficial to include more evaluative comments on the findings mentioned, highlighting their significance, limitations, or contradictions.
Also, it would be good to discuss the broader impact of MPX-endocrine interactions on public health and clinical practices. For instance, how could this knowledge influence diagnostic protocols, treatment approaches, or preventive measures? How clinicians could monitor endocrine functions in MPX patients more effectively? Why should this matter to clinicians, researchers, or public health policymakers?
Author Response
Dear Reviewers,
We sincerely thank you for your time and thoughtful evaluation of our manuscript titled: “Mpox Infection and Endocrine Health: Bridging the Knowledge Gap.” We highly appreciate the constructive feedback, which has greatly helped us refine and improve the manuscript’s scientific value and clarity. Following your recommendations, we have made a series of targeted revisions that enhance both the clinical relevance and methodological rigor of our review. In the following, we address each comment individually, providing a summary of the changes made and the rationale behind them.
Reviewer 2
Comment 1: “In my opinion, the abstract lacks a clear statement of the specific aim. For instance, it does not explicitly define whether the manuscript seeks to synthesize existing research, propose new hypotheses, or offer clinical management strategies.”
Response:
In response, we have revised (lines 12-17) the opening of the abstract to clearly state the specific aim and nature of the manuscript. The revised abstract now identifies the article as a narrative review and clarifies that its objective is to synthesize existing clinical, experimental, and epidemiological findings regarding endocrine complications related to Mpox infection. We also briefly outline the scope of the review, including discussion of underlying mechanisms, clinical manifestations, and implications for clinical care. We believe this adjustment enhances the clarity and focus of the abstract as suggested.
Comment 2: “It would be good to refine the introductory part: Some sentences are too long and a bit confusing. For example, "Historically confined to Central and West Africa, Mpox has demonstrated the potential for sustained human-to-human transmission," consider breaking it into two sentences or rephrasing it. "Leading to outbreaks" and "raising global health concerns" are somewhat overlapping. The phrase "risk of epidemics is not to be overlooked" is vague. Specify what kind of risks (e.g., economic burden, challenges in public health response). The introductory part should have more information about the global importance of Mpox and provide more details about its characteristics.”
Response:
In response, we have substantially revised (lines 35-77) the introduction to improve clarity, structure, and content. Long and complex sentences have been split or rephrased for better readability—for example, the sentence “Historically confined to Central and West Africa, Mpox has demonstrated the potential for sustained human-to-human transmission” has been restructured into two clearer statements. We also removed overlapping phrasing such as “leading to outbreaks” and “raising global health concerns” and replaced the vague expression “risk of epidemics is not to be overlooked” with more specific language detailing the potential for widespread outbreaks.
Furthermore, we expanded the introduction to include additional context on the global importance of Mpox, highlighting the WHO's declaration of it as a Public Health Emergency of International Concern, the scale of the 2024 outbreak in the DRC, and the disproportionate pediatric burden. We have also addressed systemic public health challenges, vaccine inequities, and the economic burden—supported by up-to-date data and appropriate references. These changes aim to provide a more comprehensive and informative overview of Mpox and its broader implications.
Comment 3: “Conclusions: I strongly recommend to improve this section since there is no discussion in the manuscript. It might be beneficial to include more evaluative comments on the findings mentioned, highlighting their significance, limitations, or contradictions. Also, it would be good to discuss the broader impact of MPX-endocrine interactions on public health and clinical practices. For instance, how could this knowledge influence diagnostic protocols, treatment approaches, or preventive measures? How clinicians could monitor endocrine functions in MPX patients more effectively? Why should this matter to clinicians, researchers, or public health policymakers?”
Response:
In response, we have incorporated (lines 360- 404) evaluative commentary on the main findings, emphasizing their clinical significance and noting key limitations, such as the lack of robust human data and the need for prospective endocrine evaluation. Beyond summarizing results, the conclusion now explores the broader impact of MPX–endocrine interactions on public health and clinical care. It discusses how this emerging knowledge could influence diagnostic protocols—by encouraging symptom-based hormonal screening—and treatment approaches, particularly regarding the cautious use of steroids and consideration of selective glucocorticoid receptor modulators. The importance of integrating endocrine endpoints into MPX surveillance systems and post-infection monitoring is also highlighted, along with the value of public and professional education. These additions clarify why endocrine involvement in MPX matters not only to clinicians managing individual patients, but also to researchers investigating viral pathophysiology and public health policymakers shaping outbreak responses. In this way, the revised conclusion addresses the reviewer’s request for a more analytical, clinically relevant, and forward-looking synthesis of the manuscript’s findings.